

# The 21st-century wetting inhibits growing surface ozone in Northwestern China

**Authors:** Xiaodong Zhang[1,*], Yu Yan[2], Ning Zhang[2], Wenpeng Wang[3], Huabing Suo[2], Xiaohu Jian[1], Chao Wang[2], Haibo Ma[2], Hong Gao[2,*], Zhaoli Yang[2], Tao Huang[2], Jianmin Ma[1]

**Affirmations:**

[1] Laboratory for Earth Surface Processes, College of Urban and Environmental Sciences, Peking University, Beijing, 100871, P. R. China

[2] Key Laboratory for Environmental Pollution Prediction and Control, Gansu Province, College of Earth and Environmental Sciences, Lanzhou University, Lanzhou, 730000, P.R. China

[3] College of Atmospheric Sciences, Lanzhou University, Lanzhou, 730000, P.R. China

[*] **Corresponding author:** Xiaodong Zhang, email: zhangxd2020@pku.edu.cn; Hong Gao, email: honggao@lzu.edu.cn

**ABSTRACT:**

Previous studies have shown that surface air temperature (SAT) facilitates the formation of surface ozone ($O_3$), while relative humidity (RH) often inhibits ozone generation. However, the degree to which $O_3$ may respond simultaneously to rising SATs and RH due to climate change remains less understood. We conducted extensive atmospheric chemistry model scenario simulations to investigate the impacts of long-term trends of humidification and warming on summer $O_3$ concentrations in Northwestern China (NW) from 1998 to 2017, a period during which this region experienced both warming and the most significant wetting trend in China. We found that the summer mean $O_3$ level in NW increased by 19.9% during this time. The changes in meteorology led to a reduction in $O_3$ levels, primarily due to humidification in NW, which counteracts the warming-induced $O_3$ increase. We demonstrate that the wetting trend in NW will continue from 2019 to 2030 under a shared socioeconomic pathway (SSP) in conjunction with a representative concentration pathway SSP5-RCP8.5, but will either cease or shift to drier conditions from 2019 to 2060 under the SSP2-RCP4.5. No significant responses of $O_3$ fluctuations to RH variations were observed, partly due to manual intervention under the SSP2-RCP4.5.

**Keywords:** Ozone, Wetting, Warming, Northwestern China, Future projection





## 1. Introduction

Atmospheric humidity is recognized as a crucial factor that directly and indirectly influences the formation and evolution of surface ozone (Ghazali et al., 2022; Han et al., 2011; Kerr et al., 2020). The relationship between atmospheric humidity and surface $O_3$ is complex. From a chemical standpoint, humidity can modify the rates of photochemical reactions, thereby affecting $O_3$ formation. Additionally, humidity plays a role in generating hydroxyl radicals (OH) in the atmosphere, which are vital for breaking down volatile organic compounds (VOCs) and other pollutants (Ding et al., 2023; Ma et al., 2021). These radicals can interact with $O_3$ precursors, impacting the rate of $O_3$ production. High humidity can also enhance aerosol formation, facilitating heterogeneous reactions. Depending on the specific chemical pathways involved, some of these reactions may either generate or deplete $O_3$ (Eck et al., 2020). From a physical perspective, humidity is critical for cloud and fog formation as well as precipitation. The latter promotes $O_3$ dissolution in water droplets and its removal from the atmosphere through wet deposition, particularly during precipitation events, which weakens solar radiation and disrupts the energy balance near the surface (Ding et al., 2021; Eck et al., 2020; Kleeman, 2008; Madden and Williams, 1978; Millstein and Harley, 2009).

It has been reported that $O_3$ levels increased by 30% over Northern China in summer and 50% over Eastern China in autumn during a dry year compared to a wet year, indicating that wet climate conditions do not support rising $O_3$ levels (Ding et al., 2023). Long-term ambient $O_3$ measurement data indicate that rising absolute air humidity resulted in lower near-surface $O_3$ concentrations at temperatures of 0−30°C (Belan and Savkin, 2019). Other studies examining the relationship between $O_3$ concentrations and atmospheric humidity have also identified negative correlations between $O_3$ levels and humidity (Li et al., 2022). Nevertheless, significant knowledge gaps regarding the relationship between humidity and $O_3$ persist (Li et al., 2021). Most previous studies utilized short-term or instantaneous $O_3$ concentrations to evaluate the associations between $O_3$ and atmospheric humidity (Gong et al., 2021; Li et al., 2021). Their long-term relationships still require clarification. Previous investigations of the connections between climate and $O_3$ primarily focused on the impact of air temperature, which often overshadows the effects of other meteorological variables (Wang et al., 2024). The extent to which atmospheric humidity could significantly influence the long-term $O_3$ trend remains uncertain, given that the synergistic effects of warming and wetting frequently occur in a warming climate. In this context, $O_3$ and RH trends in NW over





the past decades likely present a unique scenario to help bridge this knowledge gap. NW (Fig. S1),
located in the hinterland of the Asian-European continent, is Asia's largest arid and semi-arid
region, characterized by limited rainfall and vegetation (Koster et al., 2004; Yang et al., 2023).
Meteorological records indicate that from 1960 to 2013, precipitation in NW increased at a rate of
0.55 mm per year (Li et al., 2012; Liu et al., 2013; Yang et al., 2017). During this period, the air
temperature rose at a rate of 0.034 °C per year, exceeding the warming trends observed across all
of China (0.025 °C per year) and globally (0.013 °C per year). Although RH does not exhibit a
significant growth trend during this period, in the recent decade (2008-2017), it increased markedly
at a slope of 0.313 in NW and 0.381 in the three westernmost provinces of Xinjiang, Gansu, and
Qinghai within NW. Based on these records, NW appears to be entering a "warming and wetting"
phase, accompanied by a rise in precipitation, runoff, and lake levels (Nie et al., 2022; Peng and
Zhou, 2017; Shi et al., 2002; Yao et al., 2021).
The present study aims to address the knowledge gap in the complex and long-term
relationships between ozone pollution and atmospheric humidity, focusing on NW, where
increasing $O_3$ pollution, air temperature, and humidity have co-occurred (Peng and Zhou, 2017;
Yang et al., 2017), mainly since the early 2000s. This research provides valuable insights into the
broader context of air quality management and climate variability by enhancing our understanding
of the long-term relationship between ozone pollution and atmospheric humidity.

**2. Method and data**
**2.1 WRF-Chem Model Configuration**
Since measured $O_3$ concentration data in China is only available after 2013, we applied the
three-dimensional Weather Research and Forecasting with Chemistry (WRF-Chem) version 3.7
model  (http://www2.mmm.ucar.edu/wrf/users/wrf_files/wrfv3.7/updates-3.7.html)  to  predict
meteorology, $O_3$ concentrations, and other relevant data in Northwestern and mainland China. The
model domain spans the entire mainland China at a 20 km × 20 km resolution, extending vertically
from the surface to the lower stratospheric ozone layer at an altitude of 50 hPa with 30 staggered
vertical layers. This vertical segmentation facilitates a deepened comprehension of the $O_3$
dynamics within the region, enabling the capture of its detailed behavior throughout the





atmospheric stratifications (Li et al., 2020; Zhang et al., 2020, 2022). Detailed physical and
chemical schemes contributing to $O_3$ formation in WRF-Chem and emission input followed
previous studies (Bei et al., 2018; Bossioli et al., 2016; Liao et al., 2015; Tie et al., 2009; Wild et
al., 2000) and are presented in Section S1 in the Supplement and Tables S1-S2. The soil NO, $NO_x$,
and HONO emissions have been considered a necessary process contributing to surface ozone
formation in recent years (Lu et al., 2021). However, given the lack of soil emission inventories
of these reactive nitrogen species in China and the world, the present modeling study did not take
these processes into consideration. The aerosol feedback mechanisms include the direct and
indirect effects, the both are considered in WRF-Chem. The model simulations with a nudging
option in WRF were conducted from June 1$^{st}$ to August 31$^{st}$ in each summer from 1998 to 2017,
with a time-step of 30s. A spin-up time of three days was adopted before June 1$^{st}$ in each summer.
Further details are referred to Section S1 in the Supplement.
**2.2 Trend**
Considering relatively short data time series (1998-2017), the Mann-Kendall (MK) Test and Sen's
slope (El-Shaarawi and Niculescu, 1992; Kendall, 1949; Mann, 1945) were used to examine the
trend of $O_3$, near-surface air temperature (SAT) at the 2m height, and relative humidity (RH, %)
at the 2m height. The Mann-Kendall (MK) Test and Sen's Slope are both non-parametric statistical
methods commonly used for analyzing trends in time series data. The MK Test helps determine if
there is a significant trend, and Sen's slope quantifies the strength and direction of the trend (slope
value).
**2.3 $O_3$ Attribution to Meteorology and Emission**
We calculated the fraction (percentage change, %) to account for $O_3$ attribution to meteorology
and emission subject to different model scenarios. For example, we define the fraction
$F_{O3} = (C_{S3} - C_{S1}) / C_{S1} \times 100\%,$      (1)
as the attribution of $O_3$ concentration to emissions, where $C_{S1}$ and $C_{S3}$ are the summer surface $O_3$
air concentrations from the base (S1) and fixed meteorology (S3) scenarios.
**2.4 Data**



To initialize and prescribe boundary conditions in the WRF model from 1998 to 2017,
meteorological data were sourced from the ERA-Interim reanalysis from the European Centre for
Medium-Range Weather Forecast (ECMWF,
https://www.ecmwf.int/en/forecasts/datasets/reanalysisdatasets/era-interim).
Anthropogenic emissions were collected from the Emissions Database for Global
Atmospheric Research (EDGAR) version 4.3 on a spatial resolution of $0.1° \times 0.1°$, which was
available when we started this modeling investigation. A grid-mapping program implemented the
emissions into the WRF-Chem (Crippa et al., 2018; Mo et al., 2017). Biogenic emissions were
simulated using MEGAN version 2.1 (Guenther et al., 2006, 2012). A recent study suggests that
temperature can alter precursor emissions (Pfannerstill et al., 2024). However, since available
emission inventories worldwide, including the widely-used EDGAR inventory, did not account
for the influence of meteorology, typically the air temperature, on emissions but only considers
emissions from anthropogenic and natural sources, the emissions input in the present modeling
might be subject to additional uncertainties. The reduction of these uncertainties will depend on
improved inventories that take the meteorological effect into account.
We have replaced WRF land-use change (LUC) data with Chinese land use/land cover remote
sensing monitoring database provided by the Data Center for Resources and Environmental
Sciences, Chinese Academy of Sciences (RESDC) (http://www.resdc.cn), which significantly
improved China's LUC in our investigation of the effect of urbanization on $O_3$ pollution in China
(Zhang et al., 2022). The new LUC data has a spatial resolution of 1 km $\times$ 1 km, which is
extrapolated to the WRF-Chem grid cells (20 km $\times$ 20 km).
Considering that the warming and wetting trends in NW might persist in the coming decades,
we also projected future $O_3$ and meteorology in 2019, 2030, and 2060 in the WRF-Chem model
framework and simulations, respectively. We collected future anthropogenic emission data from
the Dynamic Projection Model for Emissions in China (DPEC Inventory – MEICModel,
http://meicmodel.org.cn), with a spatial resolution of $0.25° \times 0.25°$ longitude and latitude (Cheng
et al., 2021, 2023; Tong et al., 2020). The meteorological data was collected from Shared
Socioeconomic Pathways (SSP) and Representative Concentration Pathways (RCP) climate
scenarios under the CMIP6 framework (https://aims2.llnl.gov/search/cmip6/), with the MPI-
ESM1-2-HR model (Eyring et al., 2016). Compared with other models, MPI-ESM1-2-HR has
higher resolution and can simulate the complex interactions within the climate system, including





all climate scenarios (SSP2-RCP4.5 and SSP5-RCP8.5) targeted by this study, and has been widely
used in WRF simulations.

**2.5 Model Evaluations**

Rigorous comparisons between simulated $O_3$ concentrations and measured data were
conducted and presented in Figs. S2-S7 and Section S2 in the Supplement. Overall, the simulated
$O_3$ concentrations agree reasonably well with measured data, demonstrating a high level of
reliability of modeling results.

**2.6 Model Scenario Setup**

We established several model scenarios to distill signals of meteorological conditions, primarily
SAT and RH, in modeled $O_3$ concentrations. Detailed configurations are provided in Table 1. The
first scenario (S1) serves as the baseline, incorporating summer-specific meteorological variables
and emissions from 1998 to 2017 into WRF-Chem simulations, and is thus considered a realistic
scenario. In the second scenario (S2), we fixed emissions from 1998 throughout the period from
1998 to 2017, aiming to assess the significance of the meteorological effect on $O_3$ variation. In the
third scenario (S3), we fixed meteorology from 1998 during the same period, thereby emphasizing
the importance of emissions in $O_3$ variation. To evaluate the sensitivity of scenario S3, we also
conducted an additional WRF-Chem run by fixing SAT and RH only in 1998 (scenario S4) and
compared the modeled $O_3$ concentrations with those simulated in scenario S3 for 2004. The results
indicate minor differences between model scenario S3 and the results from the fixed SAT and RH
model run (Fig. S8). By calculating the differences and fractions, we can discern the contributions
of meteorology and emissions to $O_3$ formation and destruction. We also established two additional
model scenarios to project the responses of $O_3$ concentrations to future meteorology and emissions
in NW based on CMIP6 pathways SSP2-RCP4.5 and SSP5-RCP8.5, specifically for 2019, 2030,
and 2060, respectively. Detailed descriptions of these two pathways and the corresponding data
are presented in Section S3 in the Supplement.








**Table 1.** Model scenarios.

| Scenario | Simulation period | Meteorology | Emission | LUC |
|---|---|---|---|---|
| S1 | 1998-2017 | Varied | Varied | Varied |
| S2 | 1998-2017 | Varied | Fixed (1998) | Varied |
| S3 | 1998-2017 | Fixed (1998) | Varied | Varied |
| S4 | 2004 | Only fixed SAT and RH (1998) | Varied | Varied |
| S5 | 1998-2017 | Varied | Varied | Fixed (1998) |
| SSP2-4.5 | 2019, 2030, 2060 | Varied | Varied | Varied |
| SSP5-8.5 | 2019, 2030, 2060 | Varied | Varied | Varied |


**3. Results and Discussion**
**3.1 Summer Ozone Trends**
Figures 1a and 1b present MK-Test statistics and Sen's slope estimated temporal trends of summer
$O_3$ concentrations from 1998 to 2017 under scenario S1 (base scenario) and S3 with fixed
meteorology in China. In the S1 scenario (Fig. 1a), $O_3$ concentrations exhibit statistically
significant declining trends in the northernmost regions, spanning from Northeastern Xinjiang to
Northeastern China, with the most significant decline occurring in Inner Mongolia. Increasing $O_3$
concentrations are observed along Tianshan Mountain in Xinjiang, western Tibet, and central
China. The fixed meteorology scenario (S3, Fig. 1b) shows a monotonically increasing trend
across China, indicating continuously rising $O_3$ levels for the past two decades due to growing
precursor emissions, except for some grid cells in the Yangtze River Delta (YRD), the Pearl River
Delta (PRD), the Beijing-Tianjin metropolitan area, and other sporadic regions, where MK-Test
and Sen's slope identified statistically significant negative $O_3$ trends. Previous studies have also
noted the negative trends over the past decade in these regions (Hu et al., 2024; Li et al., 2022;
Wang et al., 2020; Yu et al., 2021). The Beijing-Tianjin area, YRD, and PRD are three critical
regions under China's joint efforts to prevent and control air pollution. Regional emission




mitigation in these areas may lead to varying reductions in precursor emissions and declines in $O_3$
concentrations. Overall, increasing precursor emissions have dominated the $O_3$ trend in China, as
meteorology remained constant during this period. This is evidenced by the linear trends of NOx
and NMVOC (Non-methane VOC) emissions in China from 1998 to 2017 (Fig. S9). When
comparing the MK-Test and Sen's slope results between the two scenarios model runs, the
meteorological conditions from 1998 to 2017 overshadowed the effects of emissions, primarily
contributing to the declining $O_3$ levels (negative trends) in the northernmost regions of China,
extending from Inner Mongolia to Northeastern China. Our modeling result shows that mean
summer $O_3$ concentration averaged over Inner Mongolia reached the maximum in 2007 and
slightly declined thereafter (Fig. S10). This can be partly attributed to relatively lower precursor
emissions resulting from the sparse population and limited industrial activities in this part of China,
though having a growing trend (Zhang et al., 2022, 2023).

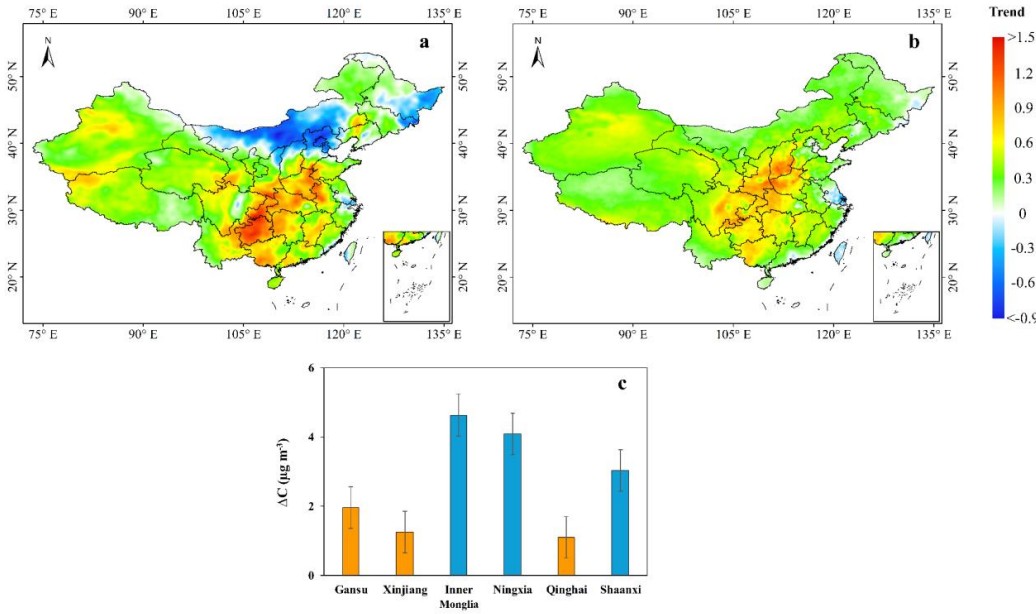


**Figure 1.** MK-Test and Sen's slope estimated summer $O_3$ concentration trend and $O_3$ attribution (fraction or
percentage change, %) under baseline (S1) and fixed meteorology (S3) scenarios from 1998 to 2017, respectively.
(a) MK-Test and Sen's slope estimated trend from the base scenario S1, the trends of concentrations are well
within the MK-Test statistically significant range at 95% confidence level in terms of |Z|>1.96 criteria; (b) same
as Fig. 1a but for fixed meteorology scenario S3; (c) mean differences of summer mean $O_3$ concentrations



averaged from 1998 to 2017 between the model scenarios 1 and 3, defined by $\overline{\Delta O_3} = \bar{O}_{3,S1} - \bar{O}_{3,S3}$, where $\bar{O}_{3,S1}$
and $\bar{O}_{3,S3}$ stand for mean summer $O_3$ concentrations averaged over 1998 to 2017, respectively. $\overline{\Delta O_3}$ in three
Westernmost provinces of NW is shown by yellow bars and blue bars for three Easternmost provinces of NW.
Error bar in Fig. 1c stands for ±1 STD.
To discern potential influences of meteorology on $O_3$ concentrations at the provincial level,
we estimated the differences of summer mean $O_3$ concentrations averaged from 1998 to 2017
between model scenarios 1 (S1) and 3 (S3), defined by $\overline{\Delta O_3} = \bar{O}_{3,S1} - \bar{O}_{3,S3}$. Figure 1c illustrates
$\overline{\Delta O_3}$ in each province of Northwestern China. Since the mean $O_3$ levels under scenario S1 were
greater than those under scenario S3 with fixed meteorology in 1998, although in some years we
also identified negative $\Delta O_3$ in the three westernmost provinces (Xinjiang, Gansu, and Qinghai),
we would expect positive $\overline{\Delta O_3}$, as shown in Fig. 1c. In this context, the values of $\overline{\Delta O_3}$ indicate the
extent of meteorological influence on $O_3$ concentrations. Specifically, the smaller $\overline{\Delta O_3}$, the
stronger meteorological influences, as a smaller $\overline{\Delta O_3}$ suggests that $O_3$ levels with varying
meteorology (S1) are closer to those with fixed meteorology in 1998 (S3). We highlighted the
three provinces with smaller $\overline{\Delta O_3}$ (<2 µg m$^{-3}$) using yellow bars, all located in Westernmost NW,
indicating a weaker increase in $O_3$ concentrations in these three westernmost provinces compared
to others, aligning well with the RH trends in these provinces, as compared to the other three
provinces located in the east of NW (Inner Mongolia, Ningxia, and Shaanxi), as mentioned in the
Introduction. As shown in Fig. S11, modeled concentration fractions under the S2 model scenario
with fixed emissions in 1998 in the three westernmost provinces of NW are negative, indicating
that meteorology tended to inhibit the increase in summer ozone concentrations, particularly in
Xinjiang, Gansu, and Qinghai, confirming the result in Fig. 1c. In the other three provinces to the
east of NW, we observe larger $\overline{\Delta O_3}$ values, suggesting that meteorology, along with precursor
emissions, tended to enhance $O_3$ levels from 1998 to 2017. Consequently, we would expect that
meteorology favored $O_3$ growth in these three eastern provinces of NW.
**3.2 Associations of summer ozone with wetting and warming**

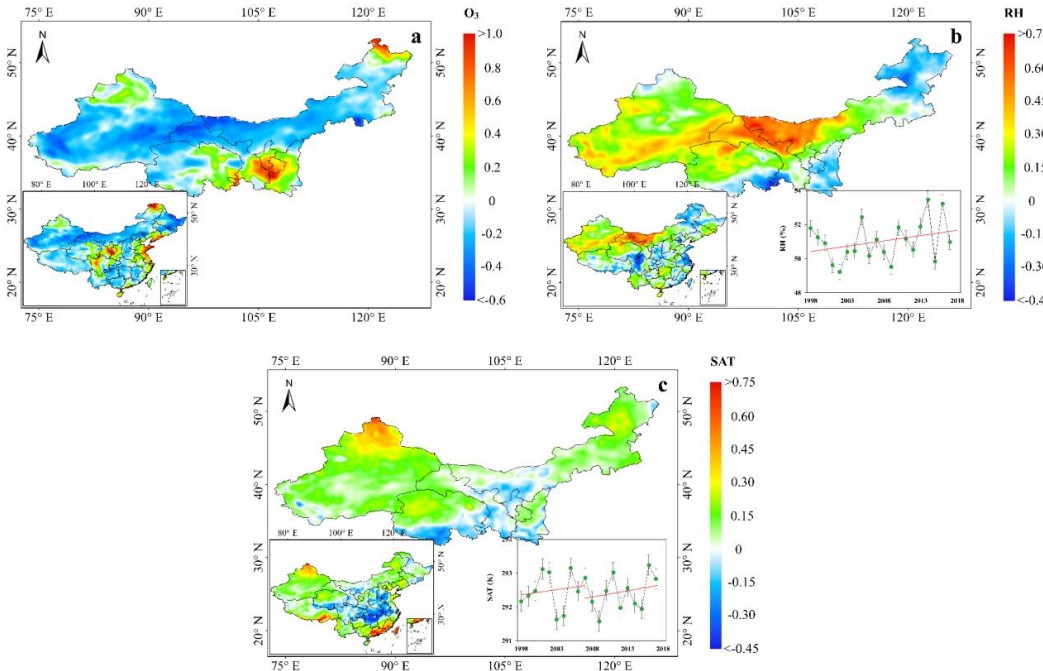


**Figure 2.** (a) MK-Test statistics for the trends of summer $O_3$ concentrations from 1998 to 2017 subject to model scenario 2 (S2) with fixed emissions in 1998, (b) MK-Test statistics for summer RH from 1998 to 2017, the line chart on the low-right corner illustrates summer RH in NW from 1998 to 2017 under scenario 2, (c) MK-Test statistics for trend of summer SATs from 1998 to 2017, the line chart shows summer SATs averaged over Northwest China from 1998 to 2017, under scenario 2. Contour figures on the low left corner of Figs. 2a-2c illustrates MK-Test statistics for $O_3$ trends under fixed emission (model scenario 2) and modeled RH and SAT trends in mainland China from 1998 to 2017. The trends are well within the statistically significant range at 95% confidence level in terms of |Z|>1.96 criteria.

Figure 2a illustrates the MK-Test statistics of gridded summer $O_3$ concentrations from 1998 to 2017 subject to the model scenario S2 with fixed emissions and annually varied meteorology, aiming to highlight the effect of meteorology on $O_3$ evolution. A notable decline in summer $O_3$ ozone concentrations is evident in most areas of mainland China, except for the Northern China Plain, central China, the Sichuan Basin, and the Guanzhong Plain (inner figure on the left-low corner of Fig. 2a). The declining $O_3$ trends are also evident in most regions across NW with the most substantial decrease occurring in west Gansu, northwest Inner Mongolia, and Xinjiang. With fixed emission, such an $O_3$ falling trend is induced primarily by meteorology (climate). Considering that air temperature and humidity are two key meteorological factors influencing $O_3$ fluctuations, we present the MK-Test statistics of the trends of summer relative humidity (RH, %)





at the 2 m height in Northwestern and mainland China (illustrated in the lower-left corner of Fig.
2b), along with their annual variations depicted in a line chart in the lower-right corner of Fig. 2b.
We also present the surface air temperature (SAT, K) in Northwestern and mainland China in a
line chart in the lower-right corner of Fig. 2c. Compared to many places across Eastern and Central
China where RH has negative or drying trends (left-low corner of Fig. 2b), RH in NW showed a
marked surge, suggesting pronounced wetting, as reported before (Peng and Zhou, 2017). This can
also be seen in the annual changes in the mean RH averaged over NW (the line chart on the right-
low corner of Fig. 2b) from 1998 to 2017, dominated by an overall upward RH trend with a mean
growth rate of 1.9% per year. Likewise, NW also experienced warming during the same period, as
shown in Fig. 2c, particularly in the western part of NW, including Xinjiang, western Gansu
Province, and Qinghai Province. NW has been undergoing one of the most significant warming
trends in the nation. Between 2005 and 2017, NW, particularly the Xinjiang region, has warmed
by over 0.5 K with a rate of 0.04 K/yr. In contrast, eastern and Central China show a cooling trend.
The rising SAT trend and falling $O_3$ trend are in contrast to the common knowledge that increasing
SATs should enhance $O_3$ levels. The result thus implies that SATs did not dominate the $O_3$ trend
in NW.



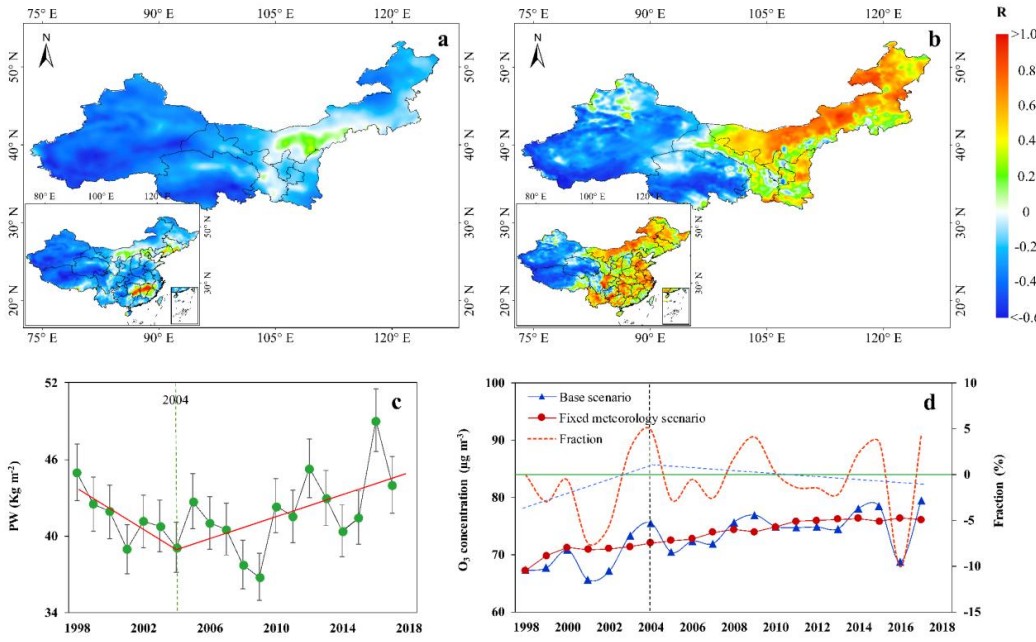

**Figure 3.** (a) Pearson correlation (R) between summer $O_3$ concentration and RH from 1998 to 2017 under S2 scenario, (b) R between summer OH radical concentration and $O_3$ concentration from 1998 to 2017 under S2 scenario, (c) summer surface precipitable water (PW) in the bounded area of NW from 1998 to 2017, error bar stands for ±1 STD and red solid line stands for linear trend, (d) summer $O_3$ concentrations under Scenarios 1 (dark blue solid line with triangles) and 3 (red solid line with circles) scaled on the left-Y axis, and their fractions $F_{O3} = (C_{S1} - C_{S3}) / C_{S3} \times 100\%$ (red dashed line) in three Westernmost provinces of HW (Xinjiang, Qinghai, and Gansu), where $C_{S1}$ and $C_{S3}$ are the summer (June to August) surface $O_3$ air concentrations from the base (S1) and fixed meteorology (S3) scenarios, scaled on the right-Y axis, the blue dashed line in Fig. 3d is a linear fitting line from 1998 to 2004 and 2004 to 2017, respectively. The green solid line represents the zero-value line of the fraction.

As mentioned earlier, since rising temperatures favor $O_3$ formation while increasing humidity is detrimental to $O_3$ growth, it is intriguing to investigate which of these two meteorological factors contributes more to the $O_3$ trend in NW. We first calculated linear correlation coefficients between RH and $O_3$ concentrations modeled from the S2 scenario (fixed emissions) from 1998 to 2017 across Northwestern and mainland China (bottom left corner of Fig. 3a). Strong negative correlations are evident in most regions of Xinjiang, Qinghai, and Western Gansu. These negative correlations indicate declining $O_3$ concentrations with rising RH. To highlight the influences of RH and precipitation on summer $O_3$ variation, Fig. 3d illustrates modeled mean summer $O_3$ concentrations averaged over three Westernmost provinces of HW (Xinjiang, Qinghai, and Gansu)





under the base scenario S1 with annually varying summer meteorology and emissions, and S3 with
fixed meteorology in 1998 and annually varying emissions averaged over NW, along with the
concentration fractions (%, red dashed line) between the two scenarios from 1998 to 2017. $O_3$
concentrations exhibit the same increasing trend primarily driven by rising precursor emissions.
However, the S1 scenario shows more significant annual fluctuations (solid dark blue line in Fig.
3d) compared to the S3 scenario (solid red line in Fig. 3d). These stronger fluctuations can be
attributed to the influence of meteorology. Since scenarios 1 and 3 shared the same emissions and
chemistry, the concentration fractions quantify the contributions of meteorology to the $O_3$ trend
alone. As shown, the fractions increased from 1998 to 2004, decreased thereafter, and turned
negative since 2009 (the red dashed lines), indicating that meteorology has led to declining $O_3$
concentrations since 2005 in NW. Among the two key meteorological factors influencing $O_3$
formation, increasing SAT (Fig. 2c) tends to elevate $O_3$ levels, which contradicts the decreasing
and negative $O_3$ fractions. Consequently, RH likely plays a more significant role in the $O_3$ trend in
NW, as rising RH aligns well with the decreasing and negative $O_3$ fractions.

Figure 3c displays the mean summer surface precipitable water (PW, kg/m2) over the region

bounded between 75 – 100°E in Northwest China from 1998 to 2017, collected from the NCAR
reanalysis (https://psl.noaa.gov/cgi-bin/data/composites/printpage.pl). One can identify a slow
decrease in PW from 1998 to 2004 and a rapidly growing PW thereafter. Since warming
exacerbates drought in arid and sub-arid NW (Lian et al., 2021), growing precipitation in NW
implies that increasing humidity might play a more significant role than warming in the changing
climate, which acts to reduce $O_3$ levels via precipitation washout and chemistry. A close look at
Figs. 2b, 2c, and Fig. 3a indicate that more substantial warming and wetting occurred in Xinjiang,
Gansu (including westernmost of Inner Mongolia), and Qinghai provinces. We compared the
fractions of provincial mean summer RH and summer $O_3$ concentrations from the fixed emissions
and annually varying meteorology scenario (S2) between 1998 and 2017 (Fig. 4a), defined by $f =$
$(Y_{2017} - Y_{1998}) / Y_{1998} \times 100\%$, where $f$ is the concentration or RH fractions and $Y_{1998}$ and $Y_{2017}$ stand
for $O_3$ concentrations or RH in 1998 and 2017, respectively, thereby to highlight the net effect
from meteorology. The most significant fraction of $O_3$ concentration was identified in Inner
Mongolia at 9.9%, followed by Ningxia (5.7%), Shaanxi (5.5%), Gansu (2.8%), Xinjiang (1.6%),
and Qinghai (-1.5%), respectively. Among the six provinces, Xinjiang and Gansu accounted for
smaller $O_3$ fractions, and Qinghai accounted for a negative fraction (Fig. 4a), indicating that $O_3$



levels in these three provinces inclined less significantly or even declined (Qinghai) between 1998
and 2017 than the other three provinces. Interestingly, more significant $O_3$ fractions, meaning
growing $O_3$ levels for the two decades in Inner Mongolia, Ningxia, and Shaanxi, correspond to
smaller (Ningxia) or negative (Inner Mongolia and Shaanxi) RH fractions, meaning falling RH. In
contrast, more diminutive and negative $O_3$ fractions in Xinjiang, Gansu, and Qinghai provinces
are associated well with the stronger increasing trend of summer RH by MK-test (Figs. 2b and 4a)
at the growing rate of 9.8% in Xinjiang, 8.9% in Gansu, and 9.9% in Qinghai from 1998 to 2017,
respectively. While the weak or falling summer $O_3$ levels in three provinces respond to both
growing RH and SAT trends (Fig. 2b and 2c), since growing SATs tend to enhance $O_3$
concentrations but modeled $O_3$ respond weakly to rising SAT in these three provinces, the result
demonstrates again that the wetting might inhibit $O_3$ formation and increment for the past decades
in NW.

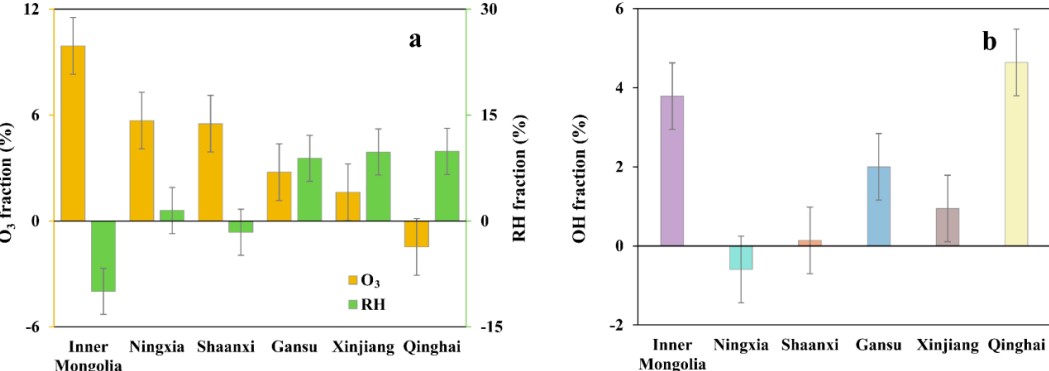


**Figure 4.** Summer RH, $O_3$, and OH radical fractions between 1998 and 2017 in six provinces of NW under the
fixed emission scenario (S2). (a) RH and $O_3$ fractions, $O_3$ fractions are scaled on the left Y-axis, and RH fractions
are scaled on the right Y-axis, (b) OH radical fractions. Error bars denote ±1 STD.
It is worthwhile to point out that model scenario 2 with fixed emissions also considers
chemistry, in which OH radicals and water vapor can interact. The primary source of OH radicals
in the atmosphere is the reaction between water vapor ($H_2O$) and excited oxygen atoms ($O(^1D)$),
which are formed by the photolysis of ozone ($O_3$) by ultraviolet (UV) radiation. The OH radicals
react with ozone to form oxygen ($O_2$) and the excited oxygen atom $O(^1D)$, which can further react
with water vapor to regenerate two OH radicals:




$O_3 + OH \rightarrow O_2 + O(^1D)$,
$O(^1D) + H_2O \rightarrow 2\ OH$.                                        (2)
This reaction does not lead to a net reduction in OH radicals but contributes to ozone removal by
initiating reactions that lead to the formation of less reactive species. Figure 3b is a correlation
diagram between modeled summer $O_3$ concentrations and OH radicals in NW and mainland China
(the left-low corner of Fig. 3b) subject to the fixed emission scenario (S2). The spatial pattern of
correlation coefficients in NW is, to some extent, similar to that between $O_3$ concentrations and
RH (Fig. 3a), showing the negative associations between OH radicals and $O_3$ concentrations,
particularly in Xinjiang and Qinghai with the most significant wetting trend, suggesting that
increasing OH radicals reduce $O_3$ levels. Figure 4b displays the fractions of provincial mean
summer OH radicals from the fixed emissions and annually varying meteorology scenario (S2)
between 1998 and 2017. Relatively larger positive fractions can be discerned in Xinjiang, Qinghai,
and Gansu, meaning increasing OH radicals in these provinces as a result of increasing RH (Eq.
1), which corresponds to higher RH fractions and lower $O_3$ fractions. An exception occurs in Inner
Mongolia, where the large positive OH radical fraction (Fig. 4b) is associated with negative RH
and large $O_3$ fractions (Fig. 4a). OH radicals show an increasing trend (Fig. S12) which agrees
with the positive correlation between OH and $O_3$ concentrations shown in Fig. 3b. This puzzling
from the modeling results needs further investigations.

There are several primary mechanisms of OH radicals leading to $O_3$ reduction, including the

reaction with VOCs to form peroxyl radicals (RO$_2$), the reaction of RO$_2$ with nitrogen oxides (NO$_x$
= NO + NO$_2$) to form nitrogen dioxide (NO$_2$) and other compounds, and photodissociation of NO$_2$.
Figures S9a and S9b show the MK test estimated NO$_x$ and NMVOC emission trends from 1998 to
2017. While increasing NO$_x$ is observed in NW (Fig. S9a), its emission levels are considerably
lower than in Eastern and Southern China (Fig. S9c). Both NMVOC emission levels and trends in
NW are significantly smaller than in Eastern and Southern China due to lower populations and
less developed economies (Fig. S9b and S9d). Hence, it is likely that the reaction of water vapor
(H$_2$O) with excited oxygen atoms (O($^1$D)) produced by the photodissociation of ozone by UV light
plays a specific role in the negative correlation between $O_3$ and OH radicals.

A multiple regression model (MRM) was established to further quantitatively assess the

contribution of RH and SAT to summer $O_3$ concentrations in NW. The MRM has been applied in



a number of investigations to quantify meteorological effects on ozone evolution in China (Li et
al., 2019, 2020, 2021; Yan et al., 2024). For SAT and RH, the MRM model can be defined as
$\quad \ln(O_3) = \alpha + \beta_1 \ln(SAT) + \beta_2 \ln(RH) + \varepsilon,$  (3)
where $\alpha$, $\beta_1$, and $\beta_2$ are model coefficients, and $\varepsilon$ is a random error, respectively. In the first instance,
we considered three provinces with significant wetting trends: Xinjiang, Gansu, and Qinghai. We
combined annual summer $O_3$, RH, and SAT from three provinces into one time series of each
dependent and independent variable from 1998 to 2017, yielding 60 samples for each variable.
Tables S4-S6 present regression coefficients and statistics of the MRM. To examine the
multicollinearity in the MRM, we estimated the variance inflation factor (VIF, Table S4-S6). The
VIFs for RH and SAT are far less than 10, indicating no multicollinearity in the model. The
contribution of RH and SAT to $O_3$ is calculated by:
$\quad \text{Contribution}_i = \dfrac{\left| \beta_i \times \frac{STD_i}{STD_\alpha} \right|}{\text{SUM}},$  (4)
where $\text{SUM} = \sum_i |\beta_i \times STD_i / STD_\alpha|$, and $\beta_i$ denotes the regression coefficient of independent
variable $i$ (namely, RH and SAT), $STD_i$ is the standard deviation of regression coefficient of
independent variable $i$, and $STD_\alpha$ represents the standard deviation of coefficient of intercept.
$\quad$ The MRM analysis reveals that RH contributes 77.5%, 59.4%, and 46.2% to $O_3$ variations in
Xinjiang, Qinghai, and Gansu. Accordingly, SAT contributes 22.5%, 40.6%, and 53.8% to $O_3$
fluctuations in these three provinces from 1998 to 2017. The results indicate that RH has become
a major contributor to long-term change in $O_3$ concentrations, overwhelming or equivalent to
SAT's contributions to the $O_3$ trend for the past two decades, likely via interactions among
increasing precipitation washout and OH radical, $O_3$ concentration, and water vapor.
$\quad$ We further assess the contributions of precipitation, RH, SAT, winds, and PBLH (planetary
boundary-layer height) to $O_3$ evolution in NW using an extended MRM model. The results confirm
that RH and SAT made the most significant contributions to long-term ozone trends than other
meteorological variables under the long-term perspective. As seen from Fig. 5, the contribution of
RH (49.5%) and SAT (47.6%) to $O_3$ evolution in NW was almost equivalent (Fig. 5a), but the
SAT (92.4%) dominated in entire mainland China (Fig. 5b). Whereas the PBLH only contributed



about 2% to O$_3$ variations because the PBLH varies on a daily and hourly basis and affects
insignificantly (summer) seasonal changes in O$_3$ concentrations.

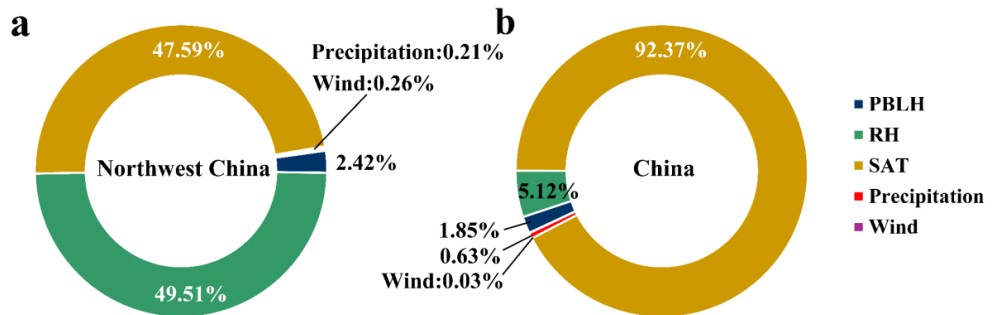


**Figure 5.** Contribution of PBLH (planetary boundary-layer height), RH, SAT, precipitation, and wind to O$_3$
evolution. (a) Northwest China, (b). Mainland China.
It should be pointed out that since the wetness also helps growing vegetation, which causes
increasing biogenic VOC emission (BVOC) and hence rising O$_3$ concentrations, the contribution
of BVOCs to O$_3$ fluctuations under increasing wetness should also be taken into consideration. To
examine this hypothesis, we designed an extra model scenario (S5) in which we used the updated
LUC data (section 2.4) and fixed LUC in 1998 in the WRF-Chem simulation and ran the model
from 1998 to 2017. Since the LUC defines vegetation coverage, the fixed LUC suggests no
vegetation change during this period. We then estimated the fractions between modeled O$_3$
concentration with fixed LUC and the baseline scenario S1 with annually-varying LUC, thereby
eliminating the effects from emissions, meteorology, and chemistry but highlighting the LUC (or
greenness) effect only. The result is illustrated in Fig. S13. As shown, while the fraction fluctuates
positively (Xinjiang) and negatively (Northern Shaanxi and Eastern Inner Mongolia), indicating
increasing and decreasing O$_3$ levels induced by LUC or greening, the magnitudes of provincial
mean concentration fractions are very small, mostly ranging from -1% to 0.3%, suggesting that
the LUC under the wetting trend in NW plays an insignificant role in O$_3$ fluctuation.
**3.3 Projected future climate impacts on summer Ozone**





Given the significant impacts of wetting on $O_3$ evolution in NW over the past two decades,
concerns have also been raised regarding $O_3$ responses to future wetting and warming. We
estimated the projected $O_3$ concentrations, RH, and SAT fractions between the baseline and fixed
meteorology scenarios under the SSP2-4.5 and SSP5-8.5 pathways (section 2.6) for the years 2019,
2030, and 2060, respectively (Figs. 6 and 7). As mentioned earlier, since both scenarios
implemented the same emissions and chemistry but the baseline scenario accounted for variable
meteorology, the fractions (or differences) between the two scenarios eliminated the effects of
emissions and chemistry, thereby isolating meteorological signals only.

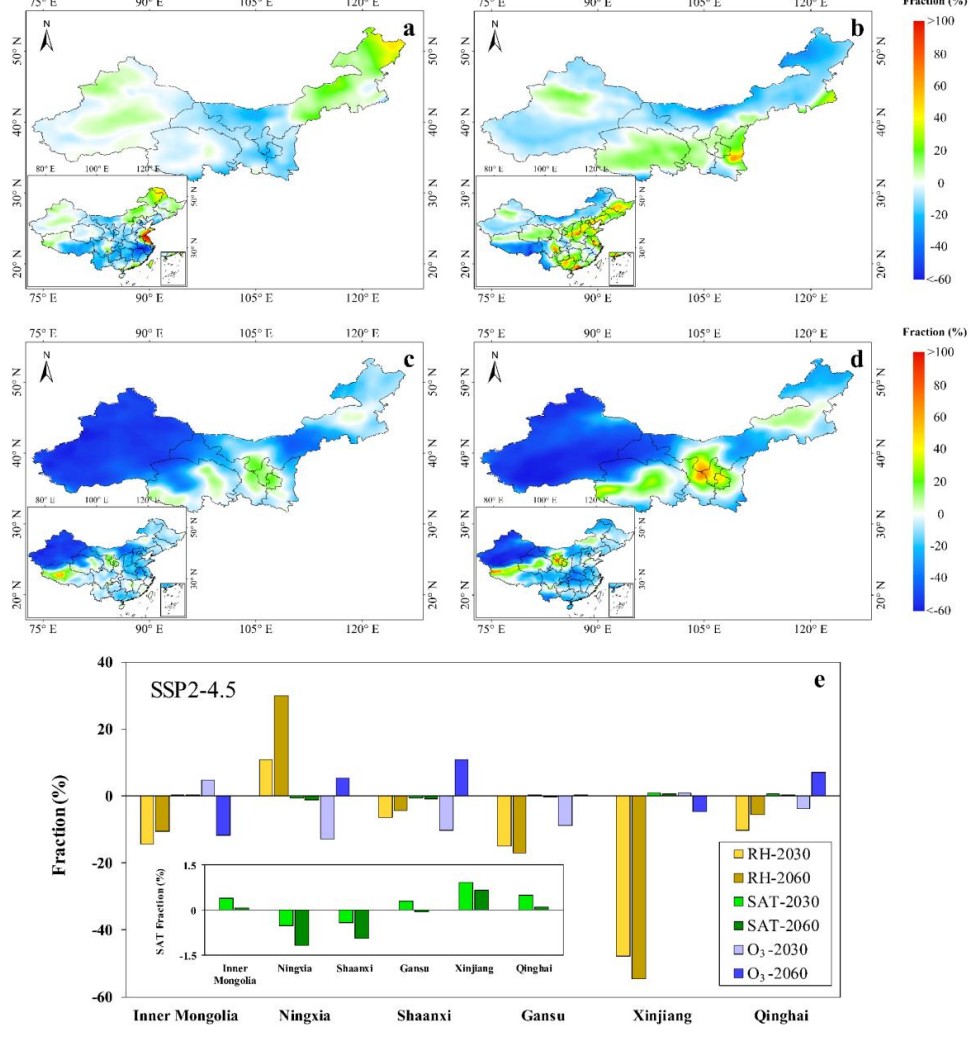






**Figure 6.** (a) Summer $O_3$ concentration fractions of 2019 to 2030 under SSP2-4.5scenario, estimated by $F_{SSP245}$ $= (C_{2030} - C_{2019}) / C_{2019} \times 100\%$, where $C_{2019}$ and $C_{2030}$ are the summer surface $O_3$ air concentrations in 2019 and 2030, respectively, (b) same as Fig. 6a but for $O_3$ fractions of 2019 to 2060, (c) same as Fig. 6a but for summer RH fractions of 2019 to 2030, (d) same as Fig. 6a but for summer RH fractions of 2019 to 2060, (e) fractions of RH, SAT, and $O_3$ of 2019 to 2030 and 2019 to 2060 (shown in the same color bar as 2030 but with oblique lines) in the six provinces of Northwestern China under the SSP2-4.5 scenario.

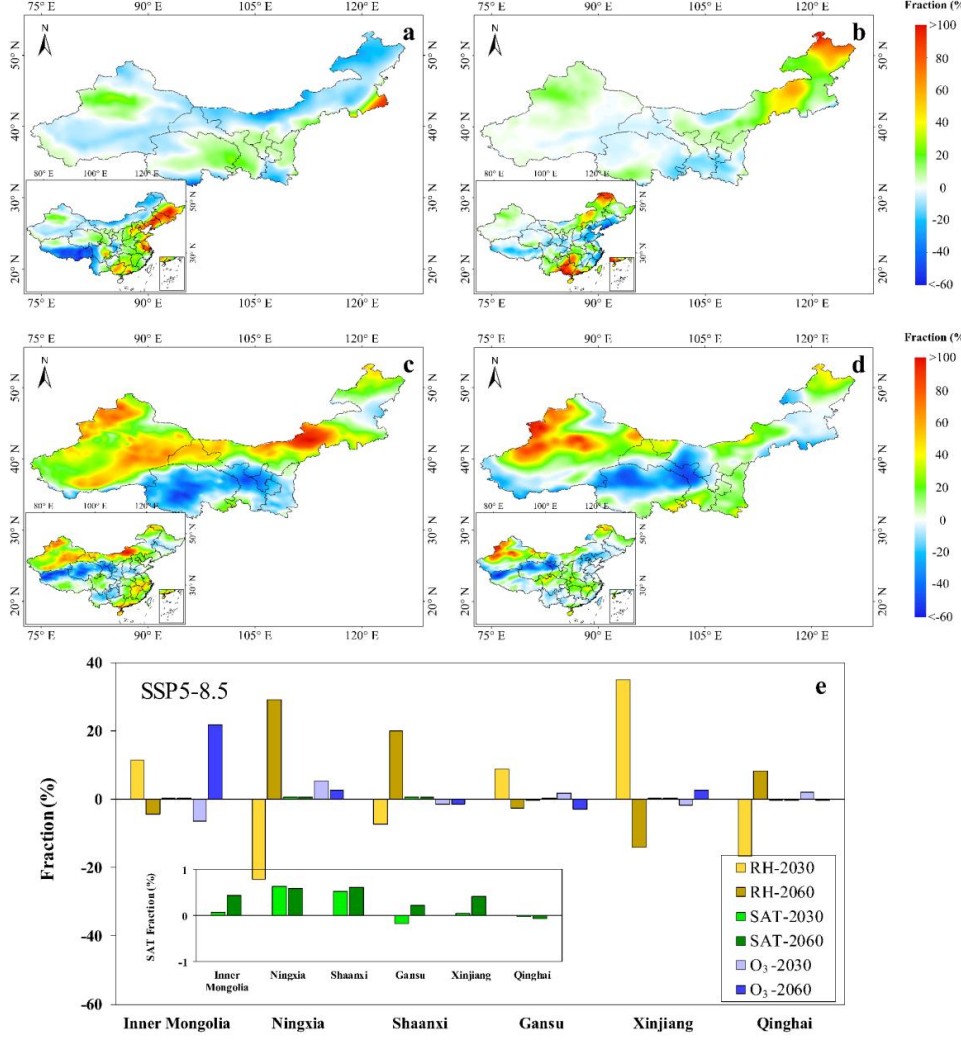

**Figure 7.** (a) Summer $O_3$ concentration fraction of 2019 to 2030 under the SSP5-8.5 scenario, estimated by $F_{SSP585} = (C_{2030} - C_{2019}) / C_{2019} \times 100\%$, where $C_{2019}$ and $C_{2030}$ are the summer surface $O_3$ air concentrations in 2019 and 2030, respectively, (b) same as Fig. 7a but for $O_3$ fraction of 2019 to 2060, (c) same as Fig. 7a but for summer RH fraction of 2019 to 2030, (d) same as Fig. 7a but for summer RH fraction of 2019 to 2060, (e)





Detailed discussions on the associations between projected O$_3$ evolution and trends of RH
and SAT under the SSP2-4.5 and SSP5-8.5 pathways from 2019 to 2030 and from 2019 to 2060
are presented in Section S4 in the Supplement and Figs. 6-7 and Fig. S14, respectively. Overall,
from 2019 to 2030, we observe a stronger negative association between O$_3$ and RH under SSP5-
8.5 than under SSP2-4.5 across NW, suggesting that climate conditions under the SSP5-8.5
scenario will more effectively suppress O$_3$ growth compared to SSP2-4.5 during this period.
However, from 2019 to 2060, the SSP5-8.5 tends to project higher O$_3$ levels than SSP2-4.5. From
a wetting trend perspective, we must determine which pathway would project a more realistic
future humidity trend in NW. China has been making significant strides in transitioning to
renewable energy sources, reducing its carbon footprint, and committing to achieve a carbon peak
before 2030 and carbon neutrality by 2060. These commitments align more closely with the
mitigation efforts outlined in SSP2-4.5, which, as manual interventions, will overshadow the
impact of climate change on O$_3$ fluctuations. There have also been debates regarding ongoing
wetting trends in NW based on CMIP6 projections under SSP5-8.5 and historical data analysis
(Huang et al., 2017). Historically, China's rapid economic growth has heavily depended on fossil
fuels, which may render SSP5-8.5 more applicable in China. The responses of O$_3$ evolution to RH
under SSP5-8.5 are more akin to those observed from 1998 to 2017 (section 3.2) than to SSP2-4.5.
In light of this, while China may achieve its carbon peak and neutrality goals, the wetting trend in
NW will likely persist in the coming decades, exacerbating continuous O$_3$ pollution in NW, in
addition to manual interventions.

**4. Conclusions**
This study conducted extensive sensitivity model simulations to explore the surface O$_3$
responses to past and future climate change in Northwestern China. The results demonstrated that
significant wetting since the 21st century in this region, particularly in the three westernmost
provinces (Xinjiang, Qinghai, and Gansu) of NW, has slowed O$_3$ growth. We show that in these
provinces, the wetting influence on O$_3$ evolution has outweighed the warming effect and played
an almost equal role to warming across the entire NW. Both factors contributed approximately 50%



to summer $O_3$ variations from 1998 to 2017, among the major meteorological factors causing
changes in $O_3$ concentrations. This finding contrasts with the dominant role of warming in the
meteorological effects on $O_3$ evolution in China. We linked the increasing OH radicals, which
interact with water vapor, to the declining $O_3$ trend. Given that humidification in NW is projected
to continue in the coming years and that the mitigation of $O_3$ precursor emissions in China will
still be implemented under the nation's air pollution control strategies, the impact of
humidification on $O_3$ trends in NW will likely become more significant. Such an impact should be
considered in future policymaking for the reduction of $O_3$ pollution in NW.

**Data availability**

ERA-Interim reanalysis data can be accessed at
https://www.ecmwf.int/en/forecasts/datasets/reanalysisdatasets/era-interim. EDGAR dataset is
available at https://edgar.jrc.ec.europa.eu/. LUC data can be accessed at http://www.resdc.cn.
DPEC Inventory is available at http://meicmodel.org.cn. The meteorological data under the
CMIP6 framework is collected from https://aims2.llnl.gov/search/cmip6/.

**Author contributions**

XZ and YY contribute equally to this article. All authors contributed to the manuscript and have
given approval of the final version. XZ and HG coordinated and supervised the project. XZ, YY,
WW and JM designed the present experiment, carried out modeling, and drafted the manuscript.
NZ, HS, XJ, CW, and HM collected the data. XZ, YY, ZY and TH analyzed simulation results.

**Competing interests**

The authors declare that they have no known competing financial interests or personal
relationships that could have appeared to influence the work reported in this paper.

**Acknowledgements**

We wish to thank the High-performance Computing Platform of Peking University to support
extensive model simulations of this study.

**Financial support**

This research was supported by the National Natural Science Foundation of China via grants
42407134, 42177351 and 41977357.

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
