# Peer review of "The 21st-century wetting inhibits growing surface ozone in Northwestern"

_EGUsphere, 2025_

## Referee Comment (RC2)

The manuscript by Zhang et al. revealed the recent wetting inhibits growing near-surface ozone in remote Northwestern China, an arid and semi-arid region located in the hinterland of the Asian-European continent based on a series of modeling experiments. The study topics of the paper is interesting, and it gives our understanding of variation of near-surface ozone under the background of regional climate change. However, the explanation of the physical and chemical processes in the simulated ozone change should be clarified. More in-depth and detailed analyses are needed to support the conclusion with major revisions:

1. Generally, Ozone variations depend both local photochemical production and regional transport. Climatologically, the contribution of foreign ozone (ozone produced outside China's troposphere) in the surface layer distributes as a "western high and eastern low" pattern over China with a large portion of near-surface ozone over Northwestern China with a large portion of near-surface ozone over Northwestern China (Li et al., 2014; Li et al., 2016). The ozone variation over Northwestern China should consider the contribution of foreign ozone.

   References

   Li, X., Liu, J., Mauzerall, D.L., Emmons, L.K., Walters, S., Horowitz, L.W., Tao, S., 2014. Effects of trans-Eurasian transport of air pollutants on surface ozone concentrations over WesternChina. J. Geophys. Res. Atmos. 119 (21), 12338–12354.

   Li, J., Yang, W., Wang, Z., Chen, H., Hu, B., Li, J., Sun, Y., Fu, P., Zhang, Y., 2016. Modeling study of surface ozone source receptor relationships in East Asia. Atmos. Res. 167, 77–88.

2. By employing the regional air quality model WRF-Chem, the recent-summer (1998-2017) ozone variations over Northwestern China were simulated. The model simulations with a nudging option in WRF were conducted from June 1st to August 31st in each summer from 1998 to 2017. Are the boundary conditions of meteorology only prescribed in the WRF model without considering the boundary conditions of chemistry in the simulation experiments? the foreign ozone contribution to the region of Northwestern China could not be simulated from 1998 to 2017. Please add a Figure of the WRF-Chem modeling domain.

3. The variations of ozone including daytime ozone formation and nighttime ozone

titration. It is only observed during daytime that surface air temperature facilitates the formation of ozone, and relative humidity inhibits ozone generation on the troposphere. Please investigate the variations of daytime ozone over Northwestern China from 1998 to 2017.

4. Both Sections 1 Introduction and 4 conclusions are too simple. Please add the reviews on the regional ozone variations over recent years in Northwestern China and the discussions on the uncertainty of this modeling study.

---

## Author Comment (AC1)

**Responses to Editor and Referee's comments**

First of all, we would like to thank the Editor and Referee for their comments and suggestions, which improved greatly the presentations and interpretations in our revised manuscript. In the revised article, we have addressed all comments and suggestions from the Editor and Referee. Our point-by-point responses to the Referee's comments are outlined below. The Referee's original comments are shown in italics and our responses are given in normal fonts.

**Referee #1**

**Comments:**

*This paper reported that the modeled decline in surface ozone levels appears closely linked to a regional wetting trend in northwestern China, in line with observations. Increased precipitation enhances the removal of ozone precursors and ozone itself through wet deposition and promotes cloud cover, which reduces photochemical ozone production by limiting sunlight. Additionally, higher humidity and wetter conditions can alter atmospheric chemistry, leading to lower ozone formation rates. Together, these factors contribute significantly to the reduction in surface ozone concentrations in northwestern China. Authors conducted extensive model sensitivity simulations to highlight falling ozone concentrations induced by rising humidity in this part of China. The results are convincing and the paper is publishable in the ACP after addressing following comments.*

**Response:** We thank the Referee's positive and encouraging comments, which help us to improve this article considerably.

*1. Figure 1 captions indicate the figure illustrates $O_3$ attribution (fraction) between the two scenarios (S1 and S3), but the figure does not show such attribution. I would understand here you mean "difference" in Fig. 1c. You can add such the attribution figure as the fraction between Fig. 1a and 1b in revised Fig. 1, which could help readers understand further the effect of meteorology on $O_3$ trend.*

**Response:** Following the Reviewer's comment, we have added a new Fig. 1c in Figure 1.

*2. Figure 2c and line 261-262, does SAT trend in scenario 2 (S2) differ from baseline scenario S1? Unless you turned on feedback simulations in WRF-Chem, SAT trends between S1 and S2 should not differ each other.*

**Response:** In the WRF-Chem simulation, we explicitly activated both aerosol feedback mechanisms (include the direct and indirect effects) and nudging option, please refer to lines 107-111.

*3. There are several objectively analyzed ozone databases providing global and China's gridded daily O$_3$ concentrations, such as MERRA-2. Authors may compare their modeled O$_3$ trend under baseline scenario (S1) with these datasets, thereby further verifying their model results.*

**Response:** Given some key uncertainties in O$_3$ concentrations near the surface in MERRA-2, particularly before 2004 (Wargan et al., 2017), MERRA-2 data was not used to compare with our modeled data. Instead, following the Reviewer's suggestion, we compared our simulated summer O$_3$ concentration trend (Fig. 1a) with CAMS (Copernicus Atmosphere Monitoring Service) global reanalysis (EAC4) archived O$_3$ date. The results show good agreements of O$_3$ trends in Eastern China but CAMS reanalysis yielded negative O$_3$ trends in a vast region of Northwestern China, except for Gansu and Shaanxi. This seems inconsistent with the background O$_3$ trend measured in Waliguan Global Atmospheric Background Station (36°17'N, 100°54'E), located in Qinghai Province, Northwestern China, where measured daily O$_3$ concentrations from 1998-2014 show a positive slope of 0.0007, implying an increasing trend. To avoid confusion, we did not present CAMS O$_3$ trends across China.

Reference:

Wargan, K., Labow, G., Frith, S., Pawson, S., Livesey, N., and Partyka, G.: Evaluation of the Ozone Fields in NASA's MERRA-2 Reanalysis, J. Clim., 30, 2961–2988, https://doi.org/10.1175/JCLI-D-16-0699.1, 2017.

---

## Author Comment (AC2)

**Responses to Editor and Referee's comments**

First of all, we would like to thank the Editor and Referee for their comments and suggestions, which improved greatly the presentations and interpretations in our revised manuscript. In the revised article, we have addressed all comments and suggestions from the Editor and Referee. Our point-by-point responses to the Referee's comments are outlined below. The Referee's original comments are shown in italics and our responses are given in normal fonts.

**Referee #2**

**Comments:**

*The manuscript by Zhang et al. revealed the recent wetting inhibits growing near-surface ozone in remote Northwestern China, an arid and semi-arid region located in the hinterland of the Asian-European continent based on a series of modeling experiments. The study topics of the paper is interesting, and it gives our understanding of variation of near-surface ozone under the background of regional climate change. However, the explanation of the physical and chemical processes in the simulated ozone change should be clarified. More in-depth and detailed analyses are needed to support the conclusion with major revisions:*

**Response:** We thank the Referee's positive and encouraging comments, which help us to improve this article considerably.

*1. Generally, Ozone variations depend both local photochemical production and regional transport. Climatologically, the contribution of foreign ozone (ozone produced outside China's troposphere) in the surface layer distributes as a "western high and eastern low" pattern over China with a large portion of near-surface ozone over Northwestern China with a large portion of near-surface ozone over Northwestern China (Li et al., 2014; Li et al., 2016). The ozone variation over Northwestern China should consider the contribution of foreign ozone.*

*References*

*Li, X., Liu, J., Mauzerall, D.L., Emmons, L.K., Walters, S., Horowitz, L.W., Tao, S., 2014. Effects of trans-Eurasian transport of air pollutants on surface ozone concentrations over Western China. J. Geophys. Res. Atmos. 119 (21), 12338–12354.*

*Li, J., Yang, W., Wang, Z., Chen, H., Hu, B., Li, J., Sun, Y., Fu, P., Zhang, Y., 2016. Modeling study of surface ozone source receptor relationships in East Asia. Atmos. Res. 167, 77–88.*

**Response:** We thank the Referee to raise this question and let us know the previous works regarding the foreign ozone contributions to $O_3$ pollution in Northwestern China.

Tropospheric $O_3$ is a primary air pollutant with strong long-range transport potential in the atmosphere among criteria pollutants. Given dominant prevailing westerly winds in the mid-latitudes of the Northern Hemisphere, Northwestern China locates in the upstream of China. One would expect this "upstream" region would receive more $O_3$ and its precursors from foreign sources in Eurasian countries to the west. This occurred particularly in early years during which China's $O_3$ levels were relatively low. Such "upstream" effect tends to diminish in recent years along with increasing $O_3$ concentrations in China in terms of the atmospheric advection equation, and is also less significant in summer (the season this study focusing on) during which atmospheric circulations are often localized, causing more localized pollution episodes, characterized by increased local emissions combined with enhanced dispersion. The latter does not favor long-range transport.

In fact, the chemical lateral boundary conditions in WRF-Chem already account for pollutant emissions from outside the model boundaries (i.e., upstream areas), as well as the effects of atmospheric transport and chemical transformations that these pollutants undergo before entering a WRF-Chem simulation domain. These boundary conditions typically come from global chemical transport model outputs (e.g., GEOS-Chem, MOZART, CAM-Chem). These global models themselves simulate emissions on a global scale (including upstream regions), chemical reactions, wet and dry deposition, and atmospheric transport processes. These datasets are usually generated or assimilated from global models and observations, also containing information on emissions and transport at the global scale. As a result, the lateral boundary conditions govern the fluxes and concentration levels of pollutants entering the model domain from external (upstream) regions. They represent the environmental or regional background concentration outside the simulation area, which already integrates emissions and transport influences from a broader region (including upstream areas).

The lateral boundary conditions of chemical species in our modeling exercise were estimated from MOZART-4 reanalysis driven fields (Emmons et al., *Geosci. Model. Dev.* 3, 43–67, 2010) on a daily basis. The ERA-Interim data with a 6-hourly time resolution and $0.7° \times 0.7°$ lat/lon spatial resolution provided by the ECMWF (European Centre for Medium-Range Weather Forecasts) were adopted as lateral boundary conditions of meteorology. Further, since the multiple model scenario simulations in our study used the same lateral boundary conditions, this effectively removed the effect of precursor emissions and atmospheric transport from Eurasian countries but focused on wetting impact on $O_3$ evolution in the internal model domain (Northwestern China).

Given that this modeling investigation did not focus on source-sink relationship but on wetting effect on $O_3$ revolution in Northwestern China, the effect of foreign emissions via chemistry lateral boundary conditions was ignored. This is done by adopting the same lateral boundary conditions in all scenario simulations, their influences were removed in the analysis of the $O_3$ differences between different model scenarios.

Following the Reviewer's comment, we have added corresponding discussions referring above points and two references recommended by the Referee in a new second paragraph of section 2.1.

In the 1st paragraph of section 2.4, we explicitly clarified the model's capability to capture regional $O_3$ transport through open chemical boundary conditions. Revised Text: "To initialize and prescribe boundary conditions in the WRF model from 1998 to 2017, meteorological data were sourced from the ERA-Interim reanalysis……. This allows the adoption of the open chemical boundary conditions to dynamically account for cross-regional $O_3$ and precursor transport, and to resolve both local photochemical production and contributions from transboundary sources, ensuring a comprehensive representation of $O_3$ dynamics."

Full citations (Emmons et al., 2010; Li et al., 2014; Li et al., 2016) are added to the reference list.

*2. By employing the regional air quality model WRF-Chem, the recent-summer (1998-2017) ozone variations over Northwestern China were simulated. The model simulations with a nudging option in WRF were conducted from June 1st to August 31st in each summer from 1998 to 2017. Are the boundary conditions of meteorology only prescribed in the WRF model without considering the boundary conditions of chemistry in the simulation experiments? the foreign ozone contribution to the region of Northwestern China could not be simulated from 1998 to 2017. Please add a Figure of the WRF-Chem modeling domain.*

**Response:** Please referred to our responses to the Referee's first comment. A new Figure S1 illustrates the WRF-Chem domain (20 km resolution), emphasizing its coverage beyond China to capture foreign ozone contributions.

*3.The variations of ozone including daytime ozone formation and nighttime ozone titration. It is only observed during daytime that surface air temperature facilitates the formation of ozone, and relative humidity inhibits ozone generation on the troposphere. Please investigate the variations of daytime ozone over Northwestern China from 1998 to 2017.*

**Response:** We agree with the Referee that ozone pollution and formation occur mainly on daytime. While daytime ozone formation and nighttime titration are critical for understanding diurnal ozone cycles, our study focuses on interannual ozone trends driven by large-scale meteorological and emission changes over decades. Although long-term ozone trends investigated in our study were constructed on an annual basis, the annual summer ozone concentrations were averaged over hourly concentrations over both daytime and nighttime.

To address the Referee's comment, we have added a new second paragraph in section 2.2. We wrote "Interannual ozone variability in Northwestern China is primarily driven by large-scale meteorological shifts (e.g., warming, humidification) and regional precursor emissions, rather than diurnal processes. Previous studies (Cooper et al., 2014; Li et al., 2020) have revealed that long-term $O_3$ trends (interannual and interdecadal scales) are robustly represented by seasonal or annual mean concentrations, as diurnal variations associated with daytime photochemistry and nighttime titration are statistically insignificant. For example, Ding et al. (2023) reported that separating daytime and nighttime $O_3$ yielded negligible differences (>95% correlation) in $O_3$ trend over a 20-year period in arid seasons in China."

Full citations for Cooper et al. (2014), Li et al. (2020), and Ding et al. (2023) are added to the reference list.

*4.Both Sections 1 Introduction and 4 conclusions are too simple. Please add the reviews on the regional ozone variations over recent years in Northwestern China and the discussions on the uncertainty of this modeling study.*

**Response:** Thank you for your constructive suggestions. We have expanded the Introduction and Conclusions sections to include a detailed review of regional ozone variations and a discussion of modeling uncertainties.

---

## Author Comment (AC3)

**Responses to Editor and Referee's comments**

First of all, we would like to thank Prof. Shao for his comments, which improved greatly the presentations and interpretations in our revised manuscript. In the revised article, we have addressed all comments from Prof. Shao. Our point-by-point responses to the Referee's comments are outlined below. The Referee's original comments are shown in italics and our responses are given in normal fonts.

**Referee #3**

**Comments:**

*Can you provide a copy of your WRF-Chem namelist?*

**Response:** The following presents the WRF-Chem namelist configuration used in this study.

```
&time_control
run_days                          = 92,
run_hours                         = 0,
run_minutes                       = 0,
run_seconds                       = 0,
start_year                        = 2017, 2000, 2000,
start_month                       = 06,    01,    01,
start_day                         = 01,    01,    24,
start_hour                        = 00,    00,    12,
start_minute                      = 00,    00,    00,
start_second                      = 00,    00,    00,
end_year                          = 2017, 2000, 2000,
end_month                         = 08,    01,    01,
end_day                           = 31,    31,    25,
end_hour                          = 18,    18,    12,
end_minute                        = 00,    00,    00,
end_second                        = 00,    00,    00,
interval_seconds                  = 21600,
input_from_file                   = .true.,.true.,.false.,
history_interval                  = 360,   60,    60,
frames_per_outfile                = 1000, 1000, 1000,
restart                           = .false.,
restart_interval                  = 1440,
io_form_history                   = 2,
io_form_restart                   = 2,
io_form_input                     = 2,
io_form_boundary                  = 2,
auxinput6_inname                  = 'wrfbiochemi_d01',
```

```
auxinput7_inname                        = 'wrffirechemi_d<domain>',
auxinput8_inname                        = 'wrfchemi_gocart_bg_d<domain>',
auxinput12_inname                       = 'wrf_chem_input',
auxinput13_inname                       = 'wrfchemv_d<domain>',
auxinput5_interval_m          = 1440, 1440, 60,
auxinput7_interval_m          = 1440, 1440, 60,
auxinput8_interval_m          = 1440, 1440, 60,
auxinput13_interval_m         = 1440, 1440, 60,
io_form_auxinput2             = 2,
io_form_auxinput5             = 2,
io_form_auxinput6             = 0,
io_form_auxinput7             = 0,
io_form_auxinput8             = 0,
io_form_auxinput12            = 0,
io_form_auxinput13            = 0,
debug_level                   = 0,
auxinput1_inname                        = "met_em.d<domain>.<date>",
/

&dfi_control
/

&domains
time_step                     = 30,
time_step_fract_num          = 0,
time_step_fract_den          = 1,
max_dom                        = 1,
s_we                           = 1,      1,        1,
e_we                           = 280,      109,        94,
s_sn                           = 1,      1,        1,
e_sn                           = 230,      109,       91,
e_vert                        = 30,      30,      28,
num_metgrid_levels           = 38,
num_metgrid_soil_levels      = 4,
dx                             = 20000, 10000,    3333.33,
dy                             = 20000, 10000,    3333.33,
grid_id                       = 1,        2,         3,
parent_id                     = 0,        1,         2,
i_parent_start               = 1,       39,       30,
j_parent_start               = 1,       28,       30,
parent_grid_ratio            = 1,        3,         3,
parent_time_step_ratio       = 1,        3,         3,
p_top_requested              = 5000,
feedback                       = 1,
```

```
smooth_option                    = 0
p_top_requested                  = 5000
zap_close_levels                 = 50
interp_type                      = 1
t_extrap_type                    = 2
force_sfc_in_vinterp             = 0
use_levels_below_ground          = .true.
use_surface                      = .true.
lagrange_order                   = 1
/
sfcp_to_sfcp                     = .true.

&physics
mp_physics                       = 2,      2,      2,
progn                            = 1,      1,      1,
ra_lw_physics                    = 1,      1,      1,
ra_sw_physics                    = 2,      2,      1,
radt                             = 10,     10,     30,
sf_sfclay_physics                = 2,      2,      1,
sf_surface_physics               = 2,      2,      1,
bl_pbl_physics                   = 2,      2,      1,
bldt                             = 0,      0,      0,
cu_physics                       = 5,      5,      0,
cu_diag                          = 1,      1,      0,
cudt                             = 0,      0,      0,
ishallow                         = 0,
isfflx                           = 1,
ifsnow                           = 1,
icloud                           = 1,
surface_input_source             = 1,
num_soil_layers                  = 4,
sf_urban_physics                 = 0,      0,      0,
mp_zero_out                      = 2,
mp_zero_out_thresh               = 1.e-12
maxiens                          = 1,
maxens                           = 3,
maxens2                          = 3,
maxens3                          = 16,
ensdim                           = 144,
cu_rad_feedback                  = .true.,
/

&fdda
grid_fdda                        =1,
```

```
gfdda_inname                    = 'wrffdda_d<domain>',
gfdda_interval_m                =360,
fgdt                            =0,
if_no_pbl_nudging_uv            =0,
if_no_pbl_nudging_t             =0,
if_no_pbl_nudging_q             =0,
if_zfac_uv                      =0,
k_zfac_uv                       =10,
if_zfac_t                       =0,
k_zfac_t                        =10,
if_zfac_q                       =0,
k_zfac_q                        =10,
guv                             =0.0003,
gt                              =0.0003,
gq                              =0.0003,
if _ramping                     =1,
dtramp_min                      =60.0,
iso_form_gfdda                  =2,
 /

 &dynamics
 rk_ord                         = 3,
 w_damping                      = 1,
 diff_opt                       = 1,
 km_opt                         = 4,
 diff_6th_opt                   = 0,        0
 diff_6th_factor                = 0.12,    0.12
 base_temp                      = 290.
 damp_opt                       = 0,
 zdamp                          = 5000.,   5000.,   5000.,
 dampcoef                       = 0.01,    0.01,    0.01
 khdif                          = 0,       0,       0,
 kvdif                          = 0,       0,       0,
 non_hydrostatic                = .true., .true., .true.,
 moist_adv_opt                  = 2,       2,       0,
 scalar_adv_opt                 = 2,       2,       0,
 chem_adv_opt                   = 2,       2,       0,
 tke_adv_opt                    = 2,       2,       0,
 time_step_sound                = 4,       4,       4,
 h_mom_adv_order                = 5,       5,       5,
 v_mom_adv_order                = 3,       3,       3,
 h_sca_adv_order                = 5,       5,       5,
 v_sca_adv_order                = 3,       3,       3,
 /
```

```
&bdy_control
spec_bdy_width                    = 5,
spec_zone                          = 1,
relax_zone                         = 4,
specified                        = .true., .false., .false.,
nested                            = .false., .true., .false.,
/

&grib2
/

&namelist_quilt
nio_tasks_per_group = 0,
nio_groups = 1,
/

&chem
kemit                              = 1,
chem_opt                           = 11,          11,
bioemdt                            = 30,          30,
photdt                             = 30,          30,
chemdt                             = 5,            5,
io_style_emissions               = 1,
emiss_opt                          = 5,            5,
emiss_opt_vol                      = 0,            0,
emiss_ash_hgt                      = 20000.,
chem_in_opt                        = 0,            0,
phot_opt                           = 2,            2,
gas_drydep_opt                     = 1,            1,
aer_drydep_opt                     = 1,            1,
bio_emiss_opt                      = 3,            3,
ne_area                            = 104
dust_opt                           = 2,
dmsemis_opt                        = 1,
seas_opt                           = 2,
depo_fact                          = 0.25
gas_bc_opt                         = 1,            1,
gas_ic_opt                         = 1,            1,
aer_bc_opt                         = 1,            1,
aer_ic_opt                         = 1,            1,
gaschem_onoff                      = 1,            1,
aerchem_onoff                      = 1,            1,
wetscav_onoff                      = 0,            0,
```

```
cldchem_onoff                        = 0,          0,
vertmix_onoff                        = 1,          1,
chem_conv_tr                         = 1,           1,
conv_tr_wetscav                      = 0,          0,
conv_tr_aqchem                       = 0,           0,
biomass_burn_opt                     = 1,          1,
plumerisefire_frq                    = 120,        120,
have_bcs_chem                           = .false., .false., .false.,
have_bcs_upper                          = .false
aer_ra_feedback                      = 1,
aer_op_opt                            = 1,
opt_pars_out                          = 1,
diagnostic_chem                       = 0,
/
```

*Why only those 8 cities are selected?*

**Response:** The selection of the 8 cities for model validation was based on spatial representativeness, data availability, and coverage of diverse environmental conditions in Northwestern China.

- Arid regions: Turpan, Karamay (low rainfall, high solar radiation)

- Semi-arid regions: Lanzhou, Shizuishan (moderate industrial/urban emissions)

- High-altitude regions: Qinghai (Tibetan Plateau influence)

- Eastern transitional zones: Tongchuan, Jinan, Zhengzhou (monsoon-affected, higher anthropogenic emissions)

These cities span a latitudinal gradient (30°N–50°N) and cover key ozone-forming environments, ensuring robust evaluation of model performance across heterogeneous landscapes.

*How did you fix SAT and RH from 1998? It seems there are large differences between Fig. S8b and Fig. S8c, can you please provide some statistical results?*

**Response:** In the fixed scenario S4, only SAT and RH were replaced with 1998 values from ERA-Interim, while all other meteorological variables (e.g., wind, pressure, cloud cover) retained their original annual values from 1998 to 2017. This partial fixation isolates the impacts of SAT/RH trends but introduces discrepancies with other dynamic variables (e.g., wind-driven transport), leading to differences between scenarios (Fig. S8b vs. S8c). Such approache has been widely used in attribution studies (e.g., Li et al., 2020; Ding et al., 2023).

We conducted T-test between two data series for Figs.S8b and S8c. The two tails T-test of 0.1065 suggests no significant difference of $O_3$ concentrations under the two model scenarios over entire model domian. Large differences seem visible in the Northern China Plain (NCP) where ozone concentrations under fixed SAT and RH in 1998 seemed higher than fixed all meteorology in 1998 (S3 scenario). We further calculated the differences of the planetary boundary layer heights (PBLH) between the two scenarios (fixed RH and SAT scenario minus scenario S3). The results are illustrated new Fig. S9. Negative PBLH differences can be seen in many areas across China, including the NCP, indicating that fixed SAT and RH scenario reduced PBLH, which often associates with stronger pollution.

Corresponding discussions and new Fig. S9 have been added to revised paper (section 2.6) and SI following the Referee's comment.